# Analysis of the Clinical Impact of the BioFire FilmArray Meningitis Encephalitis Panel on Antimicrobial Use and Duration of Therapy at an Academic Medical Center

**DOI:** 10.3390/diseases10040110

**Published:** 2022-11-20

**Authors:** Kylie Markovich, Mary Joyce B. Wingler, Kayla R. Stover, Katie E. Barber, Jamie L. Wagner, David A. Cretella

**Affiliations:** 1Steward Healthcare, Fall River, MA 02721, USA; 2Medical Center, University of Mississippi, Jackson, MS 39216, USA; 3School of Pharmacy, University of Mississippi, Jackson, MS 39216, USA

**Keywords:** meningitis, meningitis/encephalitis panel, antimicrobial stewardship, diagnostic stewardship, acyclovir

## Abstract

The purpose of this study was to assess the clinical impact of the BioFire FilmArray Meningitis/Encephalitis (ME) panel on antimicrobial use and clinical outcomes. This retrospective, quasi-experiment evaluated adult and pediatric patients with suspected ME, evidenced by cerebrospinal fluid (CSF) culture. Hospital-acquired meningitis patients and patients who received antimicrobials >48 h prior to lumbar puncture were excluded. The primary endpoint was days of antimicrobial therapy pre- and post-implementation of the ME panel. Secondary endpoints included total length of stay, 30-day readmission, and individual days of antimicrobial therapy. Two hundred and sixty-four total adult and pediatric patients were included. Antimicrobial days of therapy had a median of 3 days (IQR 0–5) in the pre vs. post group with a median of 2 days (2–5) (*p* = 0.099). Days of therapy for acyclovir were significantly decreased in the post group (median 2 days [IQR 1–3] vs. 3 days [IQR 2.5–4.5], *p* = 0.0002). There were no significant differences in the secondary endpoints. Overall, implementation of the ME panel impacted the duration of antimicrobials, particularly acyclovir; however, opportunities for further education regarding antimicrobial de-escalation and utilization of the panel were identified. Antimicrobial stewardship program intervention is critical to maximize benefit of this rapid diagnostic test.

## 1. Introduction

Community acquired meningitis and encephalitis are severe, potentially fatal infections of the central nervous system (CNS) that require a timely diagnosis and rapid administration of antimicrobials [1,2,3]. Patients who contract viral or bacterial meningitis have a mortality rate of 0.5–11.7% and 10–15%, respectively [4,5]. Cerebrospinal fluid (CSF) cultures remain a critical tool in pathogen identification, but take days to finalize and are affected by receipt of empiric antimicrobials prior to the lumbar puncture (LP) [2]. Viruses such as herpes simplex virus (HSV) 1 and 2, varicella/zoster virus (VZV), and enterovirus are common pathogens in encephalitis, and can present similarly to meningitis [3,4]. These viral pathogens are not easily identified through culture and often require specialty tests performed at reference laboratories, which can take several days to return.

The BioFire^®^ FilmArray^®^ Meningitis/Encephalitis (ME) Panel is a multiplex polymerase chain reaction (PCR) test of the CSF fluid that can detect the 14 most common pathogens among all age groups in 1 h [6,7]. The sensitivity and specificity of each individual pathogen detected in this rapid diagnostic test (RDT) varies. Leber and colleagues demonstrated a sensitivity/positive percentage of agreement (PPA) of 100% for the following pathogens: *Escherichia coli* K1, *Haemophilus influenzae*, *Streptococcus pneumoniae*, human cytomegalovirus, HSV-1, HSV-2, Human Parechovirus, VZV, and *Crytococcus gattii* [8]. Two pathogens had lower sensitivities: 95.7% for Enterovirus and 85.7% for Human Herpesvirus 6 (HHV-6). The specificity or negative percentage of agreement (NPA) was 99.2% or greater for all organisms other than *Streptococcus agalactiae* [8]. In a meta-analysis of the sensitivity and specificity of the ME Panel in both adults and pediatrics, authors found similar results with a sensitivity and specificity of >90.8%. The main difference was among HSV, as there were concerns of false negative results in HSV-1/2 [9]. There is limited evidence evaluating PCR positivity after antibiotic initiation. Prior to the creation of the ME Panel, a study in 2006 by Bronska et al. found that when CSF PCR was performed prior to antibiotics versus in the presence of antibiotics, the yield only decreased from 100% to 81%, which was higher than rates reported for standard CSF culture that declined from 52% to 7% [10].

Implementation of the ME Panel into clinical practice has yielded mixed results. Some institutions have found a decreased length of stay associated with the ME Panel [11,12]. Implementation of the ME Panel has also been shown to reduce acyclovir days of therapy in several clinical pre-post analyses [12,13,14,15]. Another pre-post study performed by Walker and colleagues showed significantly fewer patients receiving overall antimicrobials such as acyclovir, vancomycin, and ampicillin, but there were not significant decreases in duration except for in the ampicillin days of therapy [16].

The purpose of this study is to evaluate how implementation of the ME panel has affected antimicrobial duration in patients with suspected meningitis/encephalitis in the absence of formal guidance for interpretation.

## 2. Materials and Methods

### 2.1. Study Setting

The University of Mississippi Medical Center (UMMC) is a 722-bed academic medical center serving as the primary tertiary referral center for the state of Mississippi. In 2016, UMMC began using the Biofire^®^ FilmArray^®^ ME panel routinely in-house for diagnosis of CNS infections (intervention). It was implemented starting on 1 August 2016 at UMMC with no official guidance or education on its use in clinical practice. The ME panel was performed daily, including weekends, and results of the ME Panel were typically available to healthcare providers within 1–2 h of receipt by the microbiology laboratory. For CSF assessment prior to the implementation of the ME panel (pre-implementation), UMMC used a send-out non-BioFire meningitis encephalitis panel that tested 7 targets, in addition to in-house individual PCR testing methods for each of the common pathogens: *Cryptococcus* spp., *Streptococcus pneumoniae*, HSV-1, HSV-2, VZV, Epstein–Barr Virus, Enterovirus, Cytomegalovirus, and JC Virus. The turnaround time for the send-out panel was roughly 3 days.

### 2.2. Study Design

This retrospective pre-post quasi-experiment was conducted between 1 August 2013–1 August 2016 (pre-implementation) and 1 August 2017–1 August 2020 (post-implementation), allowing for a one-year washout period between groups around the implementation of the Biofire^®^ FilmArray^®^ ME panel. All patients with a CSF culture result were identified via TheraDoc (Premier^®^, Charlotte, NC, USA), an electronic clinical surveillance system, and were included in the study if the patient was admitted and LP and CSF culture were performed. Participants were excluded who had antimicrobials administered >48 h prior to LP, had a polymicrobial CSF infection, or had healthcare-associated meningitis, defined as ventriculitis or meningitis in association with invasive neurosurgical procedures (post-neurosurgical meningitis) or penetrating head trauma (post-traumatic meningitis) [17]. Additionally, patients in the post-implementation group were excluded if the ME panel was not part of their CSF diagnostic workup. Participants were screened for comorbidities using the Charlson Comorbidity Index (CCI), and also for immunocompromised conditions such as prematurity during index admission, AIDS, asplenia, sickle cell disease, autoimmune disease (not including diabetes), or immunocompromising drug therapy. AIDS was defined as an absolute CD4 count less than 200 cells/mm^3^ or CD4% less than 14% at any time since diagnosis. Normal CSF white blood cell (WBC) count was defined based on normal values for each age range (0–1 years of age: 0–30/cmm, 1–4 years of age: 0–20/cmm, 5–12 years of age: 0–10/cmm, >12 years of age: 0–5/cmm).

The primary endpoint for this study was antimicrobial days of therapy (DOT), pre- and post-implementation of the ME panel, defined as “the aggregate sum of days for which any amount of a specific antimicrobial agent was administered to individual patients as documented in the electronic medical record” [18]. Secondary endpoints included total length of stay in days, 30-day readmission to the same hospital, all-cause mortality during index admission and duration of therapy for vancomycin, CSF cephalosporins (ceftriaxone, cefotaxime, ceftazidime, cefepime), ampicillin, and acyclovir. This study was approved by the Institutional Review Board.

### 2.3. Statistics

All data were screened and collected by a single investigator using REDcap electronic data capture tools [19] and downloaded into SPSS 27 (IBM, Armonk, NY, USA, 2019) for analysis. Descriptive statistics were used to summarize patient characteristics. A Mann–Whitney U test was performed for all continuous variables including duration of antimicrobials and length of stay. Fisher’s Exact test and Chi-Squared tests were used to evaluate categorical variables, as appropriate. A *p*-value of <0.05 was considered statistically significant. A sample size of 132 patients per group was needed to meet power of 95% for this study with an effect size of 0.3 and alpha of 0.05 (G*power 3.1.9.7, 2020). Patients were screened until the necessary number of participants were included who also did not meet exclusion criteria.

## 3. Results

### 3.1. Study Subjects

A total of 431 (of a possible 4112 in the pre-intervention period and 1476 in the post-intervention period) patients were screened for enrollment. Two hundred and fourteen and 170 in the pre- and post-implementation groups met inclusion criteria. Eighty-two and 38 patients, respectively, were excluded. The most common reasons for exclusion were antimicrobials within 48 h of LP, neurosurgery, or head trauma in their past medical history. A total of 264 participants, 132 in each group, were analyzed. (Figure 1).

### 3.2. Demographic Information

Demographics were similar between groups, except the post-group had more male patients and fewer patients aged 2–17 years (Table 1). Children comprised 56% of the population. All patients had at least one sign or symptom of meningitis or encephalitis at the time of LP with most commonly reported symptoms being altered mental status and headache in adults, and fever and irritability in children (Table 2). Thirty-six patients (13.6%) were considered immunocompromised with most common causes being immunosuppressive medications and AIDS. A large portion of patients (42%) had the LP performed in the Emergency Department (ED). One quarter (66 patients) did not receive antimicrobial therapy during admission.

### 3.3. Cerebrospinal Fluid Analysis

The majority of patients (70%) had a normal CSF WBC based on age (Table 3). One-hundred and fifty-six patients (61%) had a normal CSF glucose (40–70 mg/dL), and 105 patients (42%) had a normal CSF protein (15–45 mg/dL). Forty-eight patients (18.2%) had all normal values for CSF WBC, glucose, and protein. Of the patients with normal CSF WBC, 43 of the 128 (33.6%) received no antimicrobial therapy during the admission.

In the pre-group, 38 patients had a send-out meningitis panel performed and none were positive. Five patients in the pre-group had positive CSF cultures, one patient with *S. agalactiae* and the remaining four with pathogens that would not have been detected on the current ME panel, including three coagulase negative *Staphylococcus* and one diphtheroids. No viral pathogens were detected on individual send-out viral tests in the pre-group. The ME panel detected a pathogen in 18 of 132 patients in the post-group (13.6%). Two (3.7%) and 16 (20.5%) of adult and pediatric patients, respectively, had a pathogen detected. The targets positive among adults were HHV-6 and cryptococcus, while the targets positive among pediatrics were primarily viral (enterovirus (n = 8), HHV-6 (n = 4), and human parechovirus (n = 3)). One pediatric patient had *S. pneumoniae* detected on the ME panel but not detected on CSF culture. CSF cultures were positive in four of the patients in the post-group, two *Bacillus* spp., one *S. aureus*, and one coagulase negative *Staphylococcus*.

### 3.4. Primary and Secondary Endpoints

Data regarding antimicrobial treatment and clinical outcomes are summarized in Table 4. While the total antimicrobial duration of therapy was not significantly different between the pre- and post-intervention periods (*p* = 0.099), duration of acyclovir was significantly shorter in the post-intervention group (*p* = 0.0002). Clinical secondary endpoints of hospital length of stay, 30-day readmission, and mortality were similar between groups.

## 4. Discussion

Several studies have established high sensitivity and specificity for bacterial pathogens for the BioFire^®^ FilmArray^®^ ME panel [8,9]. Despite this, implementation of the ME panel at our institution did not significantly impact antimicrobial duration of therapy, except for acyclovir. Total duration of therapy in our study was 3 and 2 days in the pre- and post-groups, respectively. This suggests after implementation of the ME panel healthcare providers continued to rely on CSF culture results before making antibiotic changes, decreasing the benefits of an RDT. These results are similar to what other studies have found [15,16]. Previous studies have demonstrated mixed results regarding impact of the ME panel on clinical outcomes [11,12,13,14,15,16]. In our study, clinical outcomes including length of stay, 30-day readmission, and all-cause mortality duFring index admission were not found to be statistically different between groups.

At our institution, the ME panel was implemented with no formal guidance. While rapid diagnostics often yield rapid and actionable information, implementation of RDTs does not always lead to rapid changes in therapy. Antimicrobial stewardship programs (ASP) are optimally positioned to improve response and outcomes associated with RDTs. The best evidence for impact of ASP with RDTs is with multiplex PCR-based blood culture identification (BCID) panels [20,21]. Macvane and colleagues showed that compared with no intervention and ASP alone, ASP plus BCID decreased time to effective therapy in patients and more rapid de-escalation in patients with positive blood cultures [20]. Institutions looking to implement rapid diagnostic tests should carefully consider guidance and requirements around ordering and who will respond to results.

Diagnostic stewardship is becoming increasingly important, and our study demonstrates several opportunities to implement interventions to improve ordering of and reaction to the ME panel. First, there was a low rate of ME panel positivity and the majority of patients (70%) had a normal CSF WBC. This may indicate over-utilization of the test. There was also a large difference noted between the adult and pediatric positivity rate of the ME panel, which could imply over-utilization of the ME panel particularly in immunocompetent, adult patients. Institutions could consider only performing the ME panel in patients with pleocytosis, but certain exceptions should be allowed. Some populations in which the ME panel should be performed regardless of CSF WBC count are neonates, children, patients who are neutropenic or those who received a prolonged duration of antibiotics prior to the LP. For example, in our study 9 of the 18 patients with a positive ME panel had a CSF WBC < 10 cells/ mm^3^, all of which were children with viral pathogens identified.

Although there was no significant difference in days of therapy overall, there was a reduction of acyclovir days of therapy in this study. This mirrors findings from other studies, as previously discussed [12,13,14,15]. Interestingly, this reduction was seen despite the concern regarding sensitivity of the ME panel for HSV found in the study by Tansarli and colleagues. Given similar clinical outcomes between groups, this may be an opportunity for an additional antimicrobial stewardship intervention at centers implementing the ME panel.

An example of “low-hanging fruit” for antimicrobial stewardship that could be targeted with the ME panel is the discontinuation of vancomycin. Vancomycin is specifically added for coverage for drug-resistant *S. pneumoniae*. In one of the largest studies evaluating accuracy of the ME panel, sensitivity/positive percentage of agreement (PPA) and specificity/negative percentage of agreement (NPA) of the ME panel for *S. pneumoniae* were 100% and 99.2%, respectively [8]. Sixteen patients had *S. pneumoniae* detected on the ME panel with no false negatives identified, but seven patients were determined to have false positives. Therefore, if the LP is obtained in a timely fashion and the ME panel is negative for *S. pneumoniae*, providers should feel confident in discontinuing vancomycin immediately rather than waiting for CSF culture results.

These results demonstrate the challenge of implementing RDT. They possess a great potential with the rapid, accurate results, but translating that to clinical change continues to be difficult. Our results highlight the importance of pairing institutional policy and ASP intervention with RDT. Based on these results, it may be beneficial to limit use of the ME panel through restriction of ordering to a specific team or by use criteria. In addition, oversight by or consultation with an infectious diseases or ASP prescriber or pharmacist should be considered with the use of this RDT. Finally, ASP interventions using clinical decision support systems, included alerts in the electronic medical record, stop dates for empiric antimicrobial therapy, and/or antimicrobial timeouts could be considered to enhance the ME panel as a RDT.

Our study is not without limitations. First, 25% of patients in this study received no antimicrobial therapy despite performance of CSF diagnostic tests. A majority of these patients were being evaluated for ocular or neurological diseases, such as multiple sclerosis. Institutions should evaluate strategies to limit ME panel utilization in patients without clear evidence of CNS infection. While power was met for this study, it did include a small sample of patients from a single center. A larger sample size over a wider geographical region would enhance the generalizability of this study. Finally, the interpretation of these results is limited by the overall low rate of positivity, particularly in the adult population.

## 5. Conclusions

Implementation of the BioFire^®^ FilmArray^®^ ME Panel at a tertiary academic medical center led to a statistically significant decrease in acyclovir days of therapy, which was consistent with previous literature. There was no difference in total days of all antimicrobials or in clinical outcomes. The low overall positivity rate of suggests overutilization of the ME panel at our institution. Based on these results, it will be critical for institutions implementing the ME panel to pair it with ASP intervention or clinical guidance for interpretation of results in order to capitalize on the benefits of the rapid diagnostic test.

## Figures and Tables

**Figure 1 diseases-10-00110-f001:**
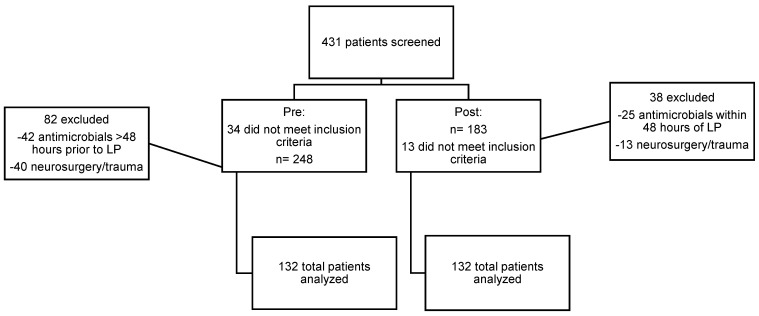
Study Flow Diagram.

**Table 1 diseases-10-00110-t001:** Demographics of study participants.

Characteristics	Pre-Group *n* = 132 (%)	Post-Group *n* = 132 (%)
Male, *n* (%) *	51 (39)	69 (52)
Age, *n* (%)		
≥50 years	27 (20)	30 (23)
≥18–49 years	35 (27)	24 (18)
2–17 years **	21 (16)	9 (7)
1 month–2 years	22 (17)	33 (25)
≤1 month	27 (20)	36 (27)
Charlson Comorbidity Score (adults only), mean (SD)	2.79 ± 2.37	3.3 ± 3.17
Penicillin allergy, *n* (%)	5 (3.8)	6 (4.5)
Immunocompromised status, *n* (%)		
Immunocompetent	113 (86)	117 (89)
Prematurity (neonates only)	2 (2)	2 (2)
HIV/AIDS	4 (3)	8 (6)
Asplenic	0 (0)	1 (1)
Sickle cell disease	3 (2.3)	1 (1)
Autoimmune disease	3 (2.3)	1 (1)
Immunosuppressive medications	10 (8)	3 (2)

* *p* = 0.035. ** *p* = 0.032. SD = standard deviation.

**Table 2 diseases-10-00110-t002:** Symptoms of meningitis/encephalitis among patients included in the pre- and post-groups at admission and divided by adult and pediatric patients.

Adult	Pre-Group *n* = 62	Post-Group *n* = 54
Symptoms, *n* (%)		
Fever	13 (21)	19 (35)
Headache	23 (37)	18 (33)
Photophobia	1 (3)	3 (6)
Nausea/vomiting	10 (16)	7 (13)
Neck stiffness	3 (5)	9 (17)
Altered mental status/confusion	23 (37)	27 (50)
Lethargy	16 (26)	20 (37)
Rash	0	1 (2)
Seizure *	5 (8)	12 (22)
**Pediatric**	**Pre-Group** ***n* = 70**	**Post-Group** ***n* = 78**
Symptoms, *n* (%)		
Fever	33 (47)	44 (56)
Headache	8 (11)	3 (4)
Photophobia	1 (1)	0
Nausea/vomiting	14 (20)	17 (22)
Neck stiffness	2 (3)	2 (3)
Altered mental status/confusion	12 (17)	7 (9)
Lethargy **	13 (19)	28 (36)
Rash	2 (3)	6 (8)
Seizure	11 (16)	17 (22)
Irritability	17 (24)	25 (32)
Lack of movement	8 (11)	10 (13)

* *p* = 0.038. ** *p* = 0.027.

**Table 3 diseases-10-00110-t003:** Cerebrospinal fluid evaluation among patients included in the pre-and post-groups.

	Pre-Group *n* = 132	Post-Group *n* = 132
Normal CSF WBC *, *n* (%)	79 (64)	90 (70)
Normal CSF glucose *, *n* (%)	77 (62)	79 (62)
Normal CSF protein *, *n* (%)	55 (44)	50 (39)
Traumatic lumbar puncture, *n* (%)	31 (25)	31 (24)

* CSF laboratory data, including WBC, glucose, and protein, are missing for some patients. All available data are reported here. WBC = white blood count; CSF = cerebrospinal fluid.

**Table 4 diseases-10-00110-t004:** Primary and Secondary Outcomes compared between included patients in the Pre-and Post-Groups.

Outcome	Pre-Group	Post-Group	*p*-Value
	*n* = 132	*n* = 132	
Total antimicrobial days of therapy, median (IQR)	3 (0–5)	2 (2–5)	0.099
Individual antimicrobial days of therapy, media (IQR)			
Vancomycin	3 (2–6.5)	3 (2–4)	0.509
CSF cephalosporin	3 (2–4)	3 (2–4)	0.384
Ampicillin	3 (3–4)	2 (1–3)	0.17
Acyclovir	3 (2.5–4.5)	2 (1–3)	0.002
Hospital length of stay in days, median (IQR)	6 (4–12)	5 (3–11.75)	0.439
30-day readmission, *n* (%)	16 (12)	15 (11)	0.848
Inpatient mortality, *n* (%)	5 (4)	1 (1)	0.107

IQR = interquartile range.

## Data Availability

Not applicable.

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
