# Peer review of "Analysis of the Clinical Impact of the BioFire FilmArray Meningitis Encephalitis Panel on Antimicrobial Use and Duration of Therapy at an Academic Medical Center"

_diseases, 2022, doi:10.3390/diseases10040110_

Round 1
Reviewer 1 Report (Previous Reviewer 2)
previous remarks have been addressed.
typo in line 260 "The low overall positivity rate of suggests overutilization of 260 the ME panel at our institution."
Reviewer 2 Report (Previous Reviewer 1)
The authors made huge edits and resolved all comments
This manuscript is a resubmission of an earlier submission. The following is a list of the peer review reports and author responses from that submission.
Round 1
Reviewer 1 Report
I appreciate the opportunity to review this interesting and well written paper. Please consider the following comments:
INTRODUCTION
The authors might cite the relevant paper entitled “Biofire FilmArray Meningitis/Encephalitis panel for the aetiological diagnosis of central nervous system infections: A systematic review and diagnostic test accuracy meta-analysis” (https://doi.org/10.1016/j.eclinm.2022.101275)
which is appropriate to this study and will help to substantiate the discussion point made on line 206 page 7.
MATERIALS AND METHODS
Study design
During the study time did you have cases of chronic meningitis or recurrent meningitis? If so, were they considered within the study population or were they eliminated?
RESULTS
Figure 1: In the pre-surgery phase, 82 subjects were excluded for not meeting the inclusion criteria, 42 of them for having started antibiotics in the 24 hours prior to the lumbar puncture.
Please correct if it was an error in the elaboration of the table or mention if they used a different inclusion criterion than the one mentioned in the text for the pre-film array phase.
Table 3: If possible, I would suggest adding another row with the number of cases with > 500 WBC/cmm.
Reviewer 2 Report
Overall
- Nice research question to examine the effect of implementing the ME panel in “real practice”. However, it is unclear how different practice was before and after implementation because no data is presented on if and by how much pathogen results were made available to the providers by implementing the ME panel. From the results I would presume that on average HSV1/2 results were 1 day quicker than before with their in house test.
- the authors conclude that providers continued to rely on CSF culture results before making antibiotic changes. This follows from their findings, but mostly for negative results. It is likely that when the ME panel is negative, that the physician decides to wait for 48-72 hours of negative culture results before stopping antibiotic treatment. Maybe the more interesting question is if physicians stop antibiotics earlier if the ME panel is positive for a viral pathogen. However the study is severely underpowered to answer this question.
Abstract
- “Days 22 of therapy for acyclovir were significantly decreased in the post group (median 3 days [IQR 2.5-4.5] 23 vs 2 days [IQR 1-3], p=0.0002)” I would suggest reversing the order.
- ASP intervention => I would suggest to write antibiotic stewardship programs in full.
Introduction
- “Patients who contract viral or bacterial meningitis have a mortality rate of 4-13%” => the quoted mortality rate of viral meningitis is unrealistically high and is not supported by the provided references.
- “Viruses 38 such as herpes simplex virus (HSV) 1 and 2, varicella/zoster virus (VZV), and enterovirus are common pathogens in encephalitis, and can present similarly to meningitis” => could the authors provide the most common viral aetiologies of encephalitis with references. VZV and enterovirus are not the first causes of encephalitis (contrary to meningitis) that come to mind.
- I am surprised that CSF pcr testing for VZV, HSV and enterovirus in the USA is mostly performed in performed at reference laboratories. Is this correct?
- “the yield decreased from 100% to 58 only 52%, however, this was still higher than rates reported for standard CSF culture” => I think the 52% cited is not correct. In table 2 from Dynamics of PCR-based diagnosis in patients with invasive meningococcal disease - ScienceDirect the PCR result is 81% after antibiotics. The 22% refers to latex agglutination. Combining both makes 52%.
Methods
- “For CSF assessment prior 77 to the implementation of the ME panel, UMMC used a send-out non-BioFire meningitis 78 encephalitis panel that tested 7 targets, in addition to in-house individual PCR testing 79 methods for each of the common pathogens: Cryptococcus spp., Streptococcus pneumoniae, 80 HSV-1, HSV-2, VZV, Epstein Barr Virus, Enterovirus, Cytomegalovirus, and JC Virus. The 81 turnaround time for the send-out panel was roughly 3 days.” => what was the turnaround time for in the in-house tests? Was in house testing done in the weekend?
- what do the authors mean by “quasi-experiment”?
- “ordinal end points including duration of antimicrobials and length of 112 stay” => does ordinal (instead of continuous) imply that duration was measured in discrete/whole days?
- “A sample size of 132 patients per group was needed to meet power of 95% for this study with an effect 115 size of 0.3 and alpha of 0.05” => what was the reason for choosing a power of 0.3 compared to the more common 0.8? Are we happy with 30% power to detect a real effect of this intervention? Especially now that we do not really find a big effect? I suspect a previous reviewer demanded a power calculation, but I doubt if this helps.
Results
- A total of 431 patients were admitted and had LP and CSF cultures performed (regardless of the result) in UMMC during the 6 year observation period, correct?
- how many patients were excluded because the ME panel was not part of their CSF diagnostic workup? Were these patients different from the included patients?
- “Demographics were similar between groups” => no statistically significant differences between groups in table 1 except sex and CCI? Could be added as subscript if the authors do not want to provide p values for all the variables. What about significant differences in the variables from table 2 and 3?
- how many patients had a pathogen detected in the pre-implementation period and which ones?
Reviewer 3 Report
Major issue:
The aim of the study was “to evaluate how implementation of the ME panel has affected antimicrobial duration in patients with suspected meningitis/encephalitis”
Study group is not well defined and described.
From table 1 some patients had normal pleocytosis. As many as 80/79 had pleocytosis of 0-10 cells same of them probably below 5 considered normal in most cases
Were patients with normal CSF pleocytosis treated with antimicrobials?
Line 227: “A majority of these patients were being evaluated for ocular or neurological diseases, such as multiple sclerosis”
Were these patients suspected for meningitis/encephalitis or LP was performed for other reasons?
I believe the aim of the study should focus on patients with confirmed meningitis only.
Is any diagnostic or therapeutic formal algorithm in use in the academic center?
Such algorithm may be crucial for the initiation and length of the antimicrobial therapy regardless of PCR results.
Minor remarks
Line 35
“Patients who contract viral or bacterial meningitis 35 have a mortality rate of 4-13% and 10-15% respectively.[4-5]”
I cannot find such high mortality in the [4] cited here. A 4-13% mortality for viral meningitis does not seem being true for me.
Figure and Tables legends are very short and not very informative.
Table 1
Pre group N=132
0-10 WBC/cmm 80 (65)
>10 WBC/cmm 43 (35)
80+43=123 Nine are missing?
Post group 79+50
Three missing?
Again: what is normal pleocytosis in the lab?